# Identification of octopaminergic neurons that modulate sleep suppression by male sex drive

Daniel R Machado[1,2,3†], Dinis JS Afonso[1,2,3†], Alexandra R Kenny[1], Arzu Öztürk-Çolak[1], Emilia H Moscato[4], Benjamin Mainwaring[4], Matthew Kayser[4], Kyunghee Koh[1*]

[1]Department of Neuroscience, the Farber Institute for Neurosciences, Kimmel Cancer Center, Thomas Jefferson University, Philadelphia, United States; [2]Life and Health Sciences Research Institute (ICVS), School of Medicine, University of Minho, Braga, Portugal; [3]ICVS/3B's - PT Government Associate Laboratory, Braga, Portugal; [4]Departments of Psychiatry and Neuroscience, Perelman School of Medicine at the University of Pennsylvania, Philadelphia, United States

**Abstract** Molecular and circuit mechanisms for balancing competing drives are not well understood. While circadian and homeostatic mechanisms generally ensure sufficient sleep at night, other pressing needs can overcome sleep drive. Here, we demonstrate that the balance between sleep and sex drives determines whether male flies sleep or court, and identify a subset of octopaminergic neurons (MS1) that regulate sleep specifically in males. When MS1 neurons are activated, isolated males sleep less, and when MS1 neurons are silenced, the normal male sleep suppression in female presence is attenuated and mating behavior is impaired. MS1 neurons do not express the sexually dimorphic FRUITLESS (FRU) transcription factor, but form male-specific contacts with FRU-expressing neurons; calcium imaging experiments reveal bidirectional functional connectivity between MS1 and FRU neurons. We propose octopaminergic MS1 neurons interact with the FRU network to mediate sleep suppression by male sex drive.

*For correspondence: kyunghee.koh@jefferson.edu

†These authors contributed equally to this work

Competing interests: The authors declare that no competing interests exist.

## Introduction

Sleep is mainly regulated by two processes: the circadian process, which controls the timing of sleep, and the homeostatic process, which modulates sleep drive based on sleep-wake history (*Borbély and Achermann, 1999*). However, because sleep is incompatible with virtually all other behaviors, sometimes it may be advantageous to forgo sleep in order to engage in other critical behaviors (*Siegel, 2012*). For example, male arctic sandpipers that sleep the least during 3 week mating periods produce the most offspring (*Lesku et al., 2012*). Elucidating the neural mechanisms underlying the choice between sleep and sex, two behaviors critical for the fitness of individuals and species, would provide valuable insights into the general problem of balancing conflicting needs.

Sleep in *Drosophila* shares many features with sleep in humans. Like humans, flies adjust their sleep behavior depending on other needs (*Griffith, 2013*). Starved flies sleep less than well-fed flies, presumably to forage for food (*Keene et al., 2010*); female flies sleep less after mating, presumably to lay eggs (*Isaac et al., 2010*); and mixed-sex groups of flies sleep less than single-sex groups, presumably to engage in sexual activities (*Liu et al., 2015*). Although several neuronal populations that regulate sleep or courtship in the fly nervous system have been identified (*Auer and Benton, 2016*; *Chakravarti et al., 2017*; *Griffith, 2013*; *Yamamoto and Koganezawa, 2013*), neural substrates underlying coordinated regulation of sleep and sexual behavior remain elusive.

**eLife digest** Most people sleep for around seven or eight hours at night, but if there is something important or interesting to do – for example, taking care of a baby, finishing a task before a deadline, or watching an entertaining movie – we may stay up late. In other words, sleep is regulated by motivational states. The drive to sleep accumulates during wakefulness and decreases during sleep. Thus sleep and other motivational drives compete to decide whether we sleep or engage in other important or interesting activities.

The idea that sleep and sex drives might compete with each other is intuitive, but had not been studied experimentally. Machado, Afonso et al. have now studied how this competition determines the behavior of male fruit flies. The presence of a female fly usually keeps a male fly awake at night. However, a male that has recently mated several times (and has low sex drive) or one that was sleep deprived (and has high sleep drive) ignores the female and goes to sleep.

Further investigation revealed a small number of previously unknown neurons (termed MS1) are required for sexual arousal to overcome the male's desire to sleep. These neurons do not belong to a circuit that is known to be important for male sexual behavior, but they do communicate with that circuit using a neurotransmitter called octopamine. This communication suppresses sleep and promotes courtship.

The next steps will be to identify the specific neurons that communicate directly with the MS1 neurons and to determine whether MS1 neurons have a direct role in regulating sex drive. Investigating these details will help us to understand more generally how competing drives influence behavioral choices.

Here we demonstrate that the balance between sleep and sex drives determine whether male flies sleep or court, and describe a newly identified neuronal group mediating sleep suppression by male sexual arousal. Earlier studies have shown that norepinephrine and its *Drosophila* counterpart octopamine act as wake-promoting signals (*Aston-Jones and Bloom, 1981*; *Carter et al., 2010*; *Crocker and Sehgal, 2008*). We found that a small number of octopaminergic neurons, which we named MS1 (Male Specific 1), regulate the decision between sleep and courtship in males. Activating MS1 neurons reduced sleep specifically in males, and silencing MS1 neurons led to decreased female-induced sleep loss and impaired mating behavior. The male-specific isoform of the FRU transcription factor FRU$^{M}$, which we will refer to as FRU for simplicity, is expressed in ~1500 neurons that range from peripheral sensory neurons to motor neurons, forming a circuit that controls courtship behavior (*Auer and Benton, 2016*; *Kimura et al., 2005*; *Manoli et al., 2005*; *Stockinger et al., 2005*; *Yamamoto and Koganezawa, 2013*). We found that MS1 neurons do not express FRU, but instead interact with the FRU neural circuit; calcium imaging experiments revealed that MS1 neurons act both upstream and downstream of FRU neurons. We propose that octopaminergic MS1 neurons communicate with the FRU courtship circuit bidirectionally to promote sexual arousal and establish a state of enhanced readiness for sustained courtship.

## Results

### Balance between sex and sleep drives determines courtship vs sleep behavior

To determine the effects of sexual stimuli on male sleep, we measured sleep in wild-type flies in different social settings: isolated male (M) or female (F) flies, and male-male (MM) or male-female (MF) pairs using multi-beam or single-beam Drosophila Activity Monitors (DAMs) (see Materials and methods). Sleep amount was markedly reduced in MF pairs relative to MM pairs (*Figure 1A,B* and *Figure 1—figure supplement 1*). As expected, isolated females exhibited reduced daytime sleep relative to isolated males, and the reduction was comparable to the daytime sleep reduction in MF relative to MM pairs (*Figure 1B* and *Figure 1—figure supplement 1B*), consistent with the possibility that the difference in daytime sleep between MF and MM pairs is largely due to female wakefulness. In contrast, nighttime sleep loss in MF relative to MM pairs is considerably greater than the

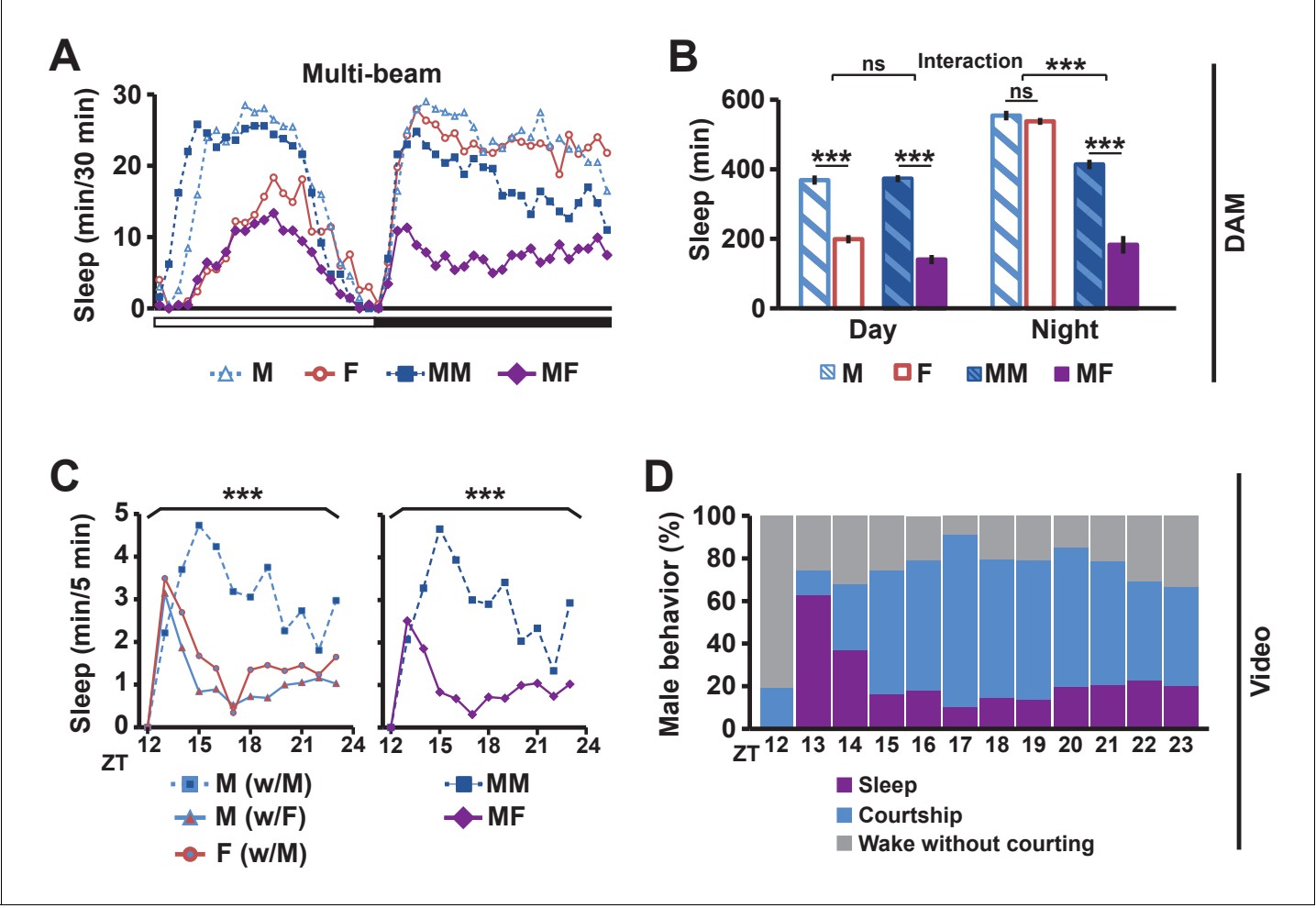

**Figure 1.** Balance between sleep drive and sex drive determines male sleep levels. (**A**) Sleep profile in 30 min intervals for wild-type (iso31) flies in isolation (M for male, F for female) or in pairs (MM for male-male, MF for male-female) using multi-beam monitors. Corresponding data using single-beam monitors are shown in *Figure 1—figure supplement 1*. In all figures, the white and black bars below the x-axis indicate light and dark periods, respectively. N = 29–44. (**B**) Daytime and nighttime sleep amount of flies shown in (**A**). (**C**) Nighttime sleep profile in 5 min bins each hour for wild-type (iso31) flies from video analysis. Sleep amount was manually scored for the first 5 min of each hour for individual males (M w/ F) or females (F w/ M) in MF pairs or individual males in MM pairs (M w/ M). Collective sleep amount of MM and MF pairs, i.e., when both flies were asleep, is presented on the right. N = 15–24. (**D**) Percentage of three types of male behavior, i.e., sleep, courtship, and wake but not courtship, during the 5 min bins shown in (**C**). In all figures, error bars represent SEM; *p<0.05; **p<0.01; ***p<0.001; ns = not significant. Two-way ANOVA (**B**) or one-way ANOVA (**C**) followed by Tukey post hoc test. Significance of the interaction between the two factors (single vs pair and presence vs absence of a female) is indicated in (**B**). Sleep amounts obtained for 5 min intervals were summed for ANOVA in (**C**) and significant differences between conditions are indicated above the brackets. Sleep amount was not significantly different between individual males and females in MF pairs (M w/ F vs F w/ M).

The following figure supplement is available for figure 1:

**Figure supplement 1.** Single-beam DAM data demonstrating reduced male sleep in the presence of females.

difference in sleep amount between isolated males and females (*Figure 1B* and *Figure 1—figure supplement 1B*), which suggests that the nighttime sleep loss in MF pairs is not simply due to the presence of another fly or sex differences in sleep amount between males and females in isolation.

To assess nighttime behavior of individual flies in MM or MF pairs, we made video recordings under infrared light, which revealed that males paired with females spent much of the night engaged in courtship (*Video 1*). To quantify this observation, we manually scored courtship and sleep-wake behavior of individual flies for the first 5 min every hour during the 12 hr dark period. Male behavior was categorized as sleep, courtship, or wake without courtship, whereas female behavior was

categorized as sleep or wake. Individual male flies slept more in MM pairs than in MF pairs (*Figure 1C*), with awake males spending most of their time courting in MF pairs (*Figure 1D*), a behavior not exhibited by either male in MM pairs. Pairs of flies tended to be awake or asleep together, and only ~2% of the time was a female awake while its male partner was asleep. As a result, sleep in a pair of flies (defined as when both flies are asleep) is a good measure of sleep in a single male fly in the pair (*Figure 1C*). These results demonstrate that males spend much of the night courting instead of sleeping when paired with females, and validate the use of DAMs to measure sleep in pairs of flies. Our data, together with previous work that employed video tracking to conclude that daily rhythms in the proximity between flies in MF pairs are driven by male sex drive (*Fujii et al., 2007*), led to the idea that male flies possess mechanisms for suppressing sleep in the presence of female flies.

If sex drive underlies female-induced male sleep loss, sexually satiated males would not exhibit sleep loss in the presence of females. To test this prediction, we employed a recently developed satiety assay (*Zhang et al., 2016*). As previously shown, male flies housed with a number of virgin females exhibited reduced courtship and copulation behaviors over a ~5 hr period (*Figure 2—figure supplement 1*). When paired with a female in a DAM, a satiated male that had been grouped with virgin females slept more than a naive male that had been grouped with other males (*Figure 2A*). Video analysis confirmed that satiated males exhibited increased sleep accompanied by decreased courtship index (*Figure 2B,C*). These data suggest that when sex drive is satisfied, the normal level of nighttime sleep drive is sufficient to allow males to sleep in the presence of females.

In a complementary experiment, we tested whether excessive sleep drive can overcome sex drive in non-satiated males by depriving them of sleep by mechanical stimulation. Whereas MF pairs slept less than MM pairs under non-deprived conditions, MF pairs slept as much as MM pairs following 6 hr of sleep deprivation (*Figure 2D*). Video analysis confirmed that sleep-deprived males slept more and courted less than non-deprived males (*Figure 2E,F*). These results demonstrate that excessive sleep drive can overcome sex drive.

We next examined how activation of the sleep-promoting dorsal fan-shaped body (dFSB) affects sleep in MF pairs. The dFSB is thought to function in the output arm of the sleep homeostat (*Donlea et al., 2014*, *Donlea et al., 2011*). We induced sleep by activating dFSB using the R23E10-Gal4 driver (*Donlea et al., 2014*) to express the bacterial sodium channel NaChBac. As expected, activation of dFSB induced sleep in isolated males (*Figure 2—figure supplement 2*). Notably, males with activated dFSB slept almost as much when paired with control females (MF[c]) as when paired with control males (MM[c]) at night (*Figure 2G*). In contrast, parental controls, i.e., males carrying either R23E10-Gal4 or UAS-*NaChBac* alone, exhibited the normal pattern of reduced nighttime sleep in MF[c] relative to MM[c] pairs (*Figure 2G*). These data provide further evidence that elevated sleep drive suppresses sexual behavior. Together, our data demonstrate that the relative strength of sleep drive and sex drive determines whether a male engages in sleep or courtship.

## MS1 neuronal activity regulates male sleep and courtship

What are the neural mechanisms underlying the decision between sleep and courtship? In an ongoing screen for neuronal populations regulating sleep in *Drosophila*, we isolated MS1-Gal4, an enhancer trap line that is associated with sexually dimorphic regulation of sleep. The Gal4 insertion (BG02822) is in an intron of the gene encoding Multidrug-Resistance like Protein 1 (MRP1). It is not known whether MRP1 plays a role in sleep or courtship. We employed the Gal4/UAS binary expression system to express the warmth-sensitive TrpA1 channel in MS1 neurons, and activated MS1 neurons by shifting the temperature from 22°C to 29°C. We found that activation of MS1 neurons led to decreased sleep in isolated

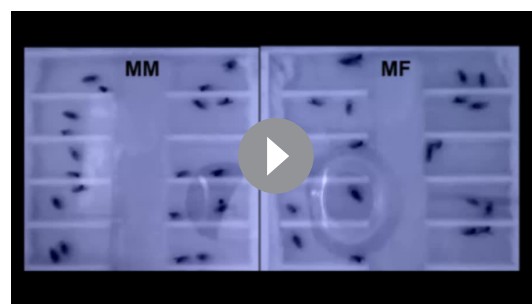

**Video 1.** Wild-type male-male (MM) and male-female (MF) pairs at ~ZT18 under infrared light. While most MM pairs slept, many males paired with females engaged in courtship behaviors including chasing and wing extension.

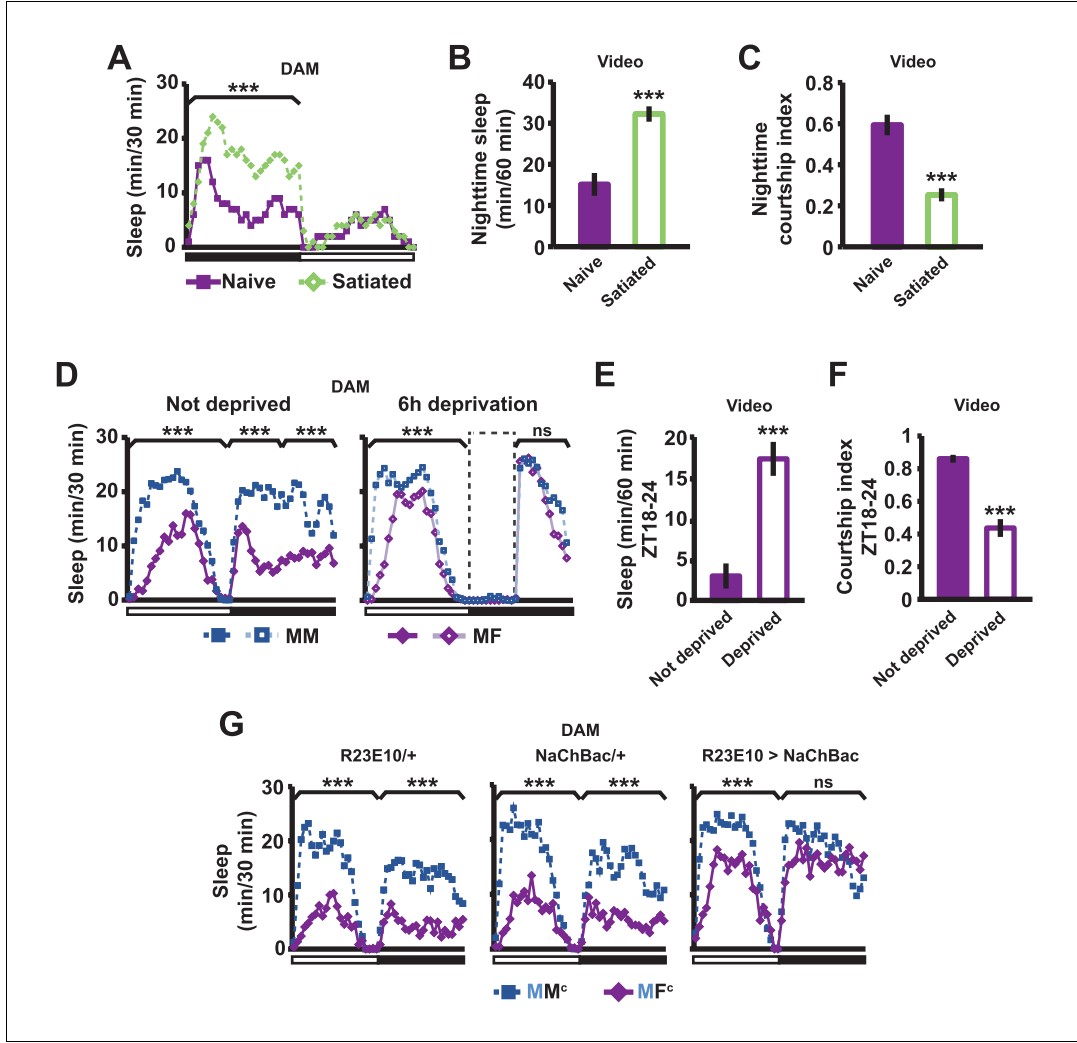

**Figure 2.** Sexual satiety and sleep deprivation attenuate female-induced sleep loss in males. (A) Sleep profile of wild-type (iso31) MF pairs including satiated or non-satiated (naive) males assayed using single-beam monitors. N = 31–33. (B–C) Sleep amount (B) and courtship index (C) of males in MF pairs from video analysis of the first 5 min every nighttime hour. N = 38–40. (D) Sleep profile of wild-type (iso31) MM or MF pairs without sleep deprivation or with 6 hr sleep deprivation in the early night measured using single-beam monitors. Dotted rectangles indicate the period of sleep deprivation by mechanical stimulation. N = 38–42. (E–F) Sleep amount (E) and courtship index (F) of males in MF pairs without sleep deprivation or with 6 hr sleep deprivation in the early night. Sleep during the first 5 min every 30 min during 6 hr after sleep deprivation (ZT18 – ZT24) was scored from videos. N = 40–48. (G) Sleep profile of 'experimental' (R23E10>NaChBac, R23E10/+, or NaChBac/+) males, paired with iso31 control females (MF$^c$) or control males (MM$^c$) measured using multi-beam monitors. N = 28–30. Student's *t* test (A–G) with Bonferroni correction (A, D, G). For statistical analysis, sleep amounts were summed and courtship index averaged over the periods indicated by the brackets above the sleep profiles or as noted on the y-axis of the bar graphs. For simplicity, in all figures involving comparison of a genotype against two parental controls, significant differences are indicated only if the experimental group differed significantly from both control groups in the same direction.

The following figure supplements are available for figure 2:

**Figure supplement 1.** Successful manipulation of sexual satiety in male flies.

**Figure supplement 2.** Activation of dFSB induces sleep in isolated males.

males, but not in isolated females (*Figure 3A*). Males with activated MS1 neurons did not exhibit male-specific behaviors, such as courtship and aggression, but instead exhibited locomotor behavior typically observed in awake flies, i.e., pacing the perimeter of the recording arena (*Video 2*). Although males with activated MS1 neurons lost about 2/3 of nighttime sleep relative to controls at 29°C, they slept no more than the control males when the temperature was returned to 22°C (*Figure 3A*), suggesting that sleep loss due to activated MS1 neurons does not lead to recovery sleep. Since the TrpA1 channels were activated for only one day during the adult stage, these results indicate that MS1 neurons function in adult male flies to promote wakefulness. Constitutive activation of MS1 neurons by NaChBac expression also resulted in male-specific sleep reduction (*Figure 3B*), demonstrating that both chronic and acute activation of MS1 neurons affect male sleep.

When we expressed tetanus toxin (TNT) in MS1 neurons to block neurotransmission, sleep in isolated males was not altered (*Figure 3C*). We hypothesized this may be because MS1 neurons become activated only under specific social contexts. Indeed, whereas silencing of MS1 neurons via TNT expression did not affect sleep in MM^c pairs (experimental males paired with control males), it led to a significant occlusion of the nighttime sleep loss in MF^c pairs (*Figure 3C*). For further analysis of male sleep and courtship behavior in MF^c pairs, we examined videos recorded under infrared light. We quantified sleep amount and courtship index for the first 5 min every nighttime hour. Males with silenced MS1 neurons slept more and courted less than parental control males (*Figure 3D,E*). These data suggest that activation of MS1 neurons is dependent on female cues and is required to keep males awake in the presence of females, presumably so that they can engage in sustained courtship.

The above experiment was conducted at night when sleep drive is high and MF pairs had been together for over a day. To further examine the role of MS1 neurons in male mating behavior, we performed courtship and copulation assays during the day immediately after a virgin female was introduced to a virgin male. Inhibition of MS1 activity by TNT expression had little effect on courtship index and copulation latency when assayed under white light conditions (*Figure 3F,G*). Since the sleep-suppressing effects of MS1 activity manipulation were stronger in the dark, we repeated the assays in infrared light during the subjective day. We found that whereas inhibition of MS1 activity by TNT expression had little effect on courtship index, it had a significant effect on copulation latency (*Figure 3F,G*). Since males with silenced MS1 neurons can successfully copulate in the light condition, it is unlikely that MS1 neuronal activity is directly required for copulation. Instead it may be required for timely progression through stages of courtship in the dark. Consistent with delayed copulation in these assays, MS1>TNT males were also less successful in mating when competing against control males (*Figure 3H*). Together, our findings suggest that MS1 neuronal activity is important for optimal mating performance in the dark, especially during the night when sleep drive is high.

## Octopamine mediates male sleep regulation by MS1 neurons

We next examined the expression pattern of MS1-Gal4 using membrane-bound GFP, and found a restricted pattern of expression in 3 to 7 neurons in the subesophageal ganglion (SOG) near the midline and ~4 pairs of neurons in the dorsal brain (*Figure 4A*). No cell bodies were detectable in the ventral nerve cord, although there were descending projections from the central brain (*Figure 4B*). Differences between male and female expression patterns were not apparent. We noticed that some of the projection patterns of the MS1 neurons in the SOG resemble those of octopaminergic neurons (*Busch et al., 2009*; *Busch and Tanimoto, 2010*). In addition, previous findings showed that sleep loss through activation of octopaminergic neurons does not result in rebound sleep (*Seidner et al., 2015*),

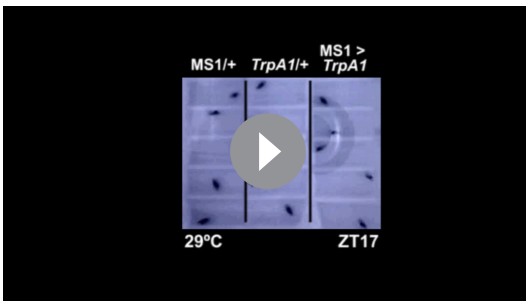

**Video 2.** Male flies with activated MS1 neurons (MS1 > *TrpA1*) and control males (MS1/+ or *TrpA1*/+) at 29°C. The flies were recorded at ~ZT17 under infrared light. MS1 activation led to pacing behavior typically observed in awake flies (note one of the control flies exhibiting the same behavior).

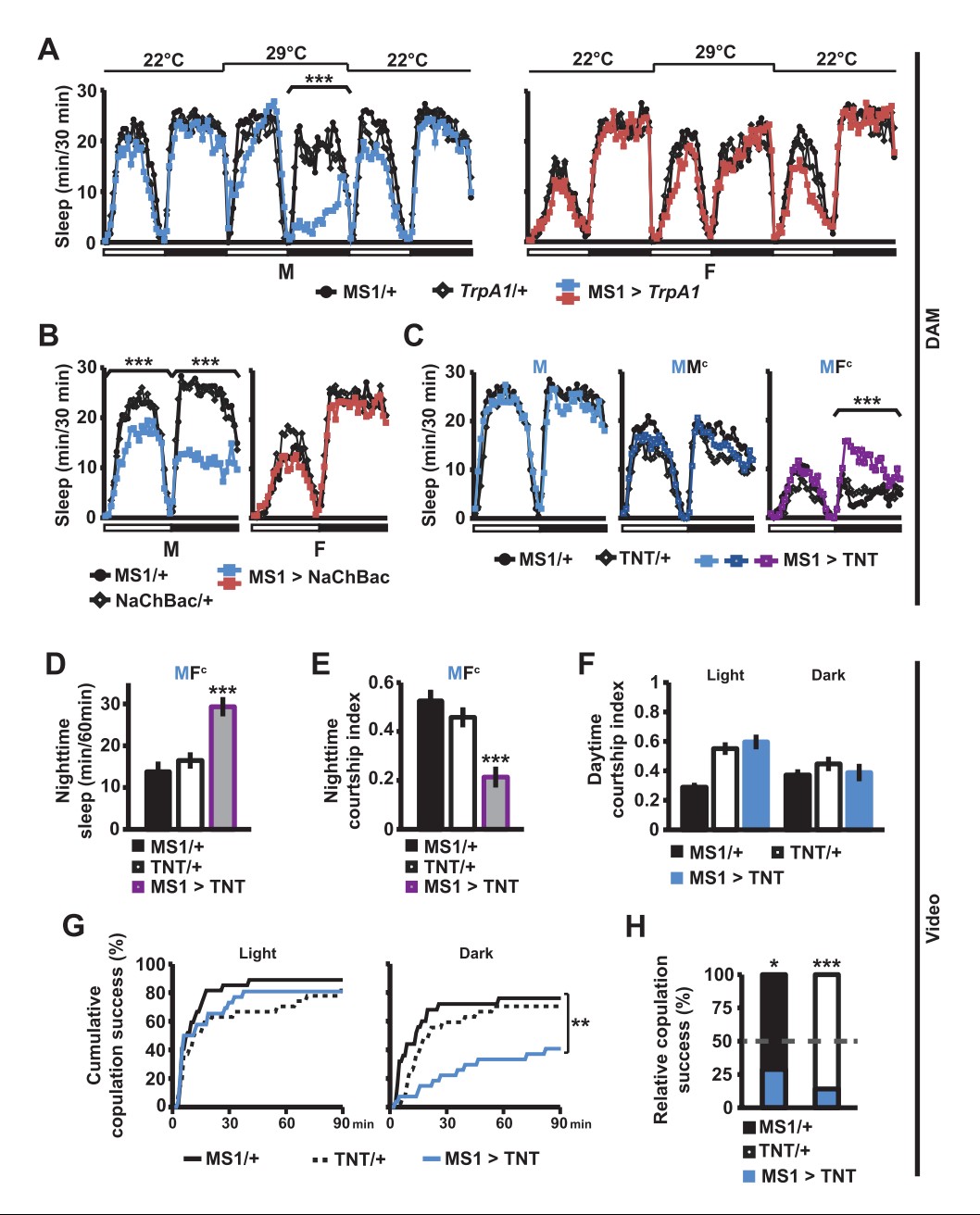

**Figure 3.** MS1 neuronal activity regulates sleep and mating behavior in males. (**A**) Sleep profile of male (left) and female (right) flies expressing the warmth-activated TrpA1 channel in MS1 neurons (MS1>TrpA1) and control flies (MS1/+ and TrpA1/+). TrpA1 was activated by temperature shift from 22°C to 29°C. N = 32–40. (**B**) Sleep profile of flies expressing the NaChBac bacterial sodium channel in MS1 neurons and parental controls. N = 30–36. (**C**) Sleep profile of males expressing tetanus toxin in MS1 neurons (MS1>TNT) and parental controls (MS1/+ and TNT/+) in isolation (M) or paired with iso31 control males (MMᶜ) or females (MFᶜ) in multi-beam monitors. N = 37–49. (**D–E**) Sleep amount (**D**) and courtship index (**E**) of males with silenced MS1 neurons (MS1>TNT) and parental controls paired with control females. Videos were manually scored for sleep and courtship index during the first 5 min bins every hour at night and values summed (**D**) or averaged (**E**) for statistical analysis. N = 26–27. (**F–G**) courtship index (**F**), and cumulative copulation success (**G**) over 90 min mating assays. Males with silenced MS1 neurons (MS1>TNT) and parental controls were assayed in dim white light (left) or infrared light (right). N = 23–26. (**H**) Percentage of successful copulation by males with silenced MS1 neurons (MS1>TNT) against MS1/+ or TNT/+ controls in 90 min competitive copulation assays under dim red light. N = 29–34. One-way ANOVA followed by Dunnett post hoc test relative to both parental controls (**A–F**); log rank test (**G**); binomial test relative to 50% (**H**).

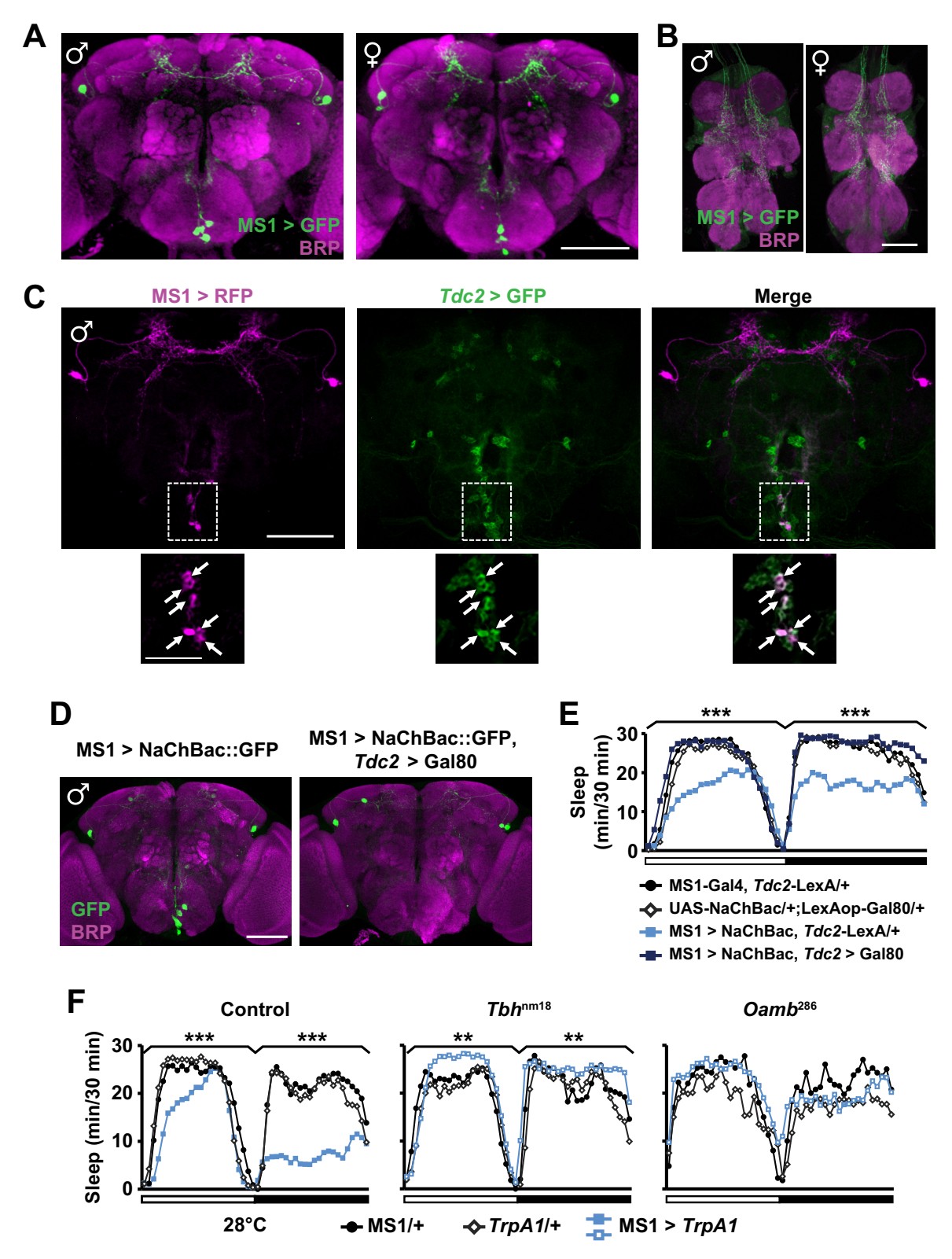

**Figure 4.** MS1 neurons in the SOG are octopaminergic. (**A**) Confocal projection of a whole-mounted MS1> CD8::GFP male (left) or female (right) adult central brain. Antibodies against GFP (green) and Bruchpilot (BRP, magenta) were used for immunostaining. Scale bar: 100 µm. (**B**) Confocal projection of the ventral nerve cord of a male or female expressing CD8::GFP under the control of MS1-Gal4. Scale bar: 50 µm. (**C**) Top: confocal projection of a male central brain expressing RFP (magenta) in MS1 neurons (using Gal4-UAS) and GFP (green) in Tdc2 neurons (using LexA-LexAOp). Scale bar: 100

*Figure 4 continued on next page*

*Figure 4 continued*

μm. Bottom: Magnified view of the SOG region indicated by the rectangle in the corresponding image in the top row. Arrows point to neurons co-expressing MS1>RFP and *Tdc2*>GFP. Scale bar: 50 μm. (D) Expression of NaChBac::GFP in all MS1 neurons (left, MS1-Gal4, *Tdc2*-LexA/UAS-*NaChBac*) and in the non-octopaminergic subset (right, MS1-Gal4, *Tdc2*-LexA/UAS-*NaChBac*; LexOp-Gal80). Expression of Gal80, a suppressor of Gal4, in TDC2 neurons removed NaChBac::GFP expression specifically in the SOG. Scale bar: 100 μm. (E) Sleep profile of male flies of the indicated genotypes. N = 75–78. (F) Sleep profile of males in which MS1 neurons were activated with TrpA1 expression at 28°C in an iso31 control (left), $Tbh^{nm18}$ mutant (middle), or $Oamb^{286}$ mutant (right) background. N = 20–165. One-way ANOVA followed by Dunnett post hoc test relative to MS1>NaChBac, Tdc2-LexA/+ flies (E) or both parental controls (F).

and light inhibits the wake-promoting effects of octopamine (*Shang et al., 2011*). These observations led us to hypothesize that some of the MS1 neurons are octopaminergic. To test this idea, we expressed RFP in MS1 neurons and GFP in TDC2-expressing octopaminergic neurons (TDC2 is an enzyme required for octopamine synthesis in the nervous system [*Cole et al., 2005*]). We found that all MS1 neurons in the SOG are octopaminergic (*Figure 4C*). Consistent with this finding, co-expression of Gal80, an inhibitor of Gal4, in TDC2 neurons removed all MS1-Gal4 activity in the SOG (*Figure 4D*).

To test whether the octopaminergic subset of MS1 neurons underlie male-specific sleep suppression, we restricted NaChBac expression to the non-octopaminergic subset of MS1 neurons using Gal80 in TDC2 neurons. Indeed, we found that activation of non-TDC2 MS1 neurons did not alter sleep in males (*Figure 4E*). Furthermore, mutations in Tyramine β-hydroxylase (Tbh), another protein involved in octopamine synthesis (*Monastirioti et al., 1996*), and OAMB, an octopamine receptor implicated in sleep regulation (*Crocker et al., 2010*), prevented the male sleep loss induced by activation of MS1 neurons (*Figure 4F*). These data show that a small number of octopaminergic neurons in the SOG mediate male-specific arousal through the OAMB receptor.

## MS1 and FRU neurons appear to be connected anatomically

Given the male-specific role of MS1 neurons in sleep regulation and the central role of the FRU circuit in the regulation of male behaviors, we employed two approaches to test whether MS1 neurons express FRU. First, we used the FLP-FRT system to restrict GFP expression to the intersection of MS1-Gal4 and *fru*-LexA (*Mellert et al., 2010*), and second, we used *fru*-LexA to express GFP in FRU neurons while simultaneously expressing RFP under the control of MS1-Gal4. We did not detect any overlap between the MS1 and FRU populations (*Figure 5A,B* and *Figure 5—figure supplement 1A*), nor did we detect DOUBLESEX (DSX), another transcription factor important for male sexual behavior, in MS1 neurons (*Figure 5C*).

Three sets of octopaminergic neurons in the SOG have been characterized previously, but MS1 neurons appear to be distinct based on 3 lines of evidence. First, 3 FRU-expressing octopaminergic neurons in the SOG underlie the decision between courtship and aggression (*Certel et al., 2010*), but MS1 neurons do not express FRU. Second, a subset of octopaminergic neurons in the SOG defined by *Tdc2*-Gal4 and *Cha*-Gal80 mediate aggression (*Zhou et al., 2008*), but the combination of MS1-Gal4 and *Cha*-Gal80 removes all expression from the SOG. Third, VPM5 is implicated in memory formation (*Burke et al., 2012*), but cell body locations and projection patterns suggest MS1 neurons do not include VPM5. Thus MS1 neurons are distinct from previously characterized octopamingeric neurons.

Interestingly, we observed a group of FRU neurons in close proximity to the MS1 neurons in the SOG (*Figure 5B*), suggesting a possible interaction between the two neuronal groups. We thus examined whether MS1 and FRU neurons can form synaptic contacts by employing a version of GRASP (GFP Reconstitution Across Synaptic Partners) in which one of the GFP fragments is fused to NEUREXIN (NRX) for synaptic targeting (*Fan et al., 2013*; *Feinberg et al., 2008*; *Gordon and Scott, 2009*). We observed that punctate reconstituted GFP (i.e., GRASP) signals in the region surrounding the esophagus were more pronounced in male brains compared with female brains (*Figure 5D,E*). GRASP signals were not observed in control flies, which demonstrates that the observed GRASP signals are not due to leaky transgene expression (*Figure 5—figure supplement 1B*). These data demonstrate sexual dimorphism in the potential contacts between MS1 and FRU neurons, which may account for the male-specific effects on sleep with activation of MS1 neurons.

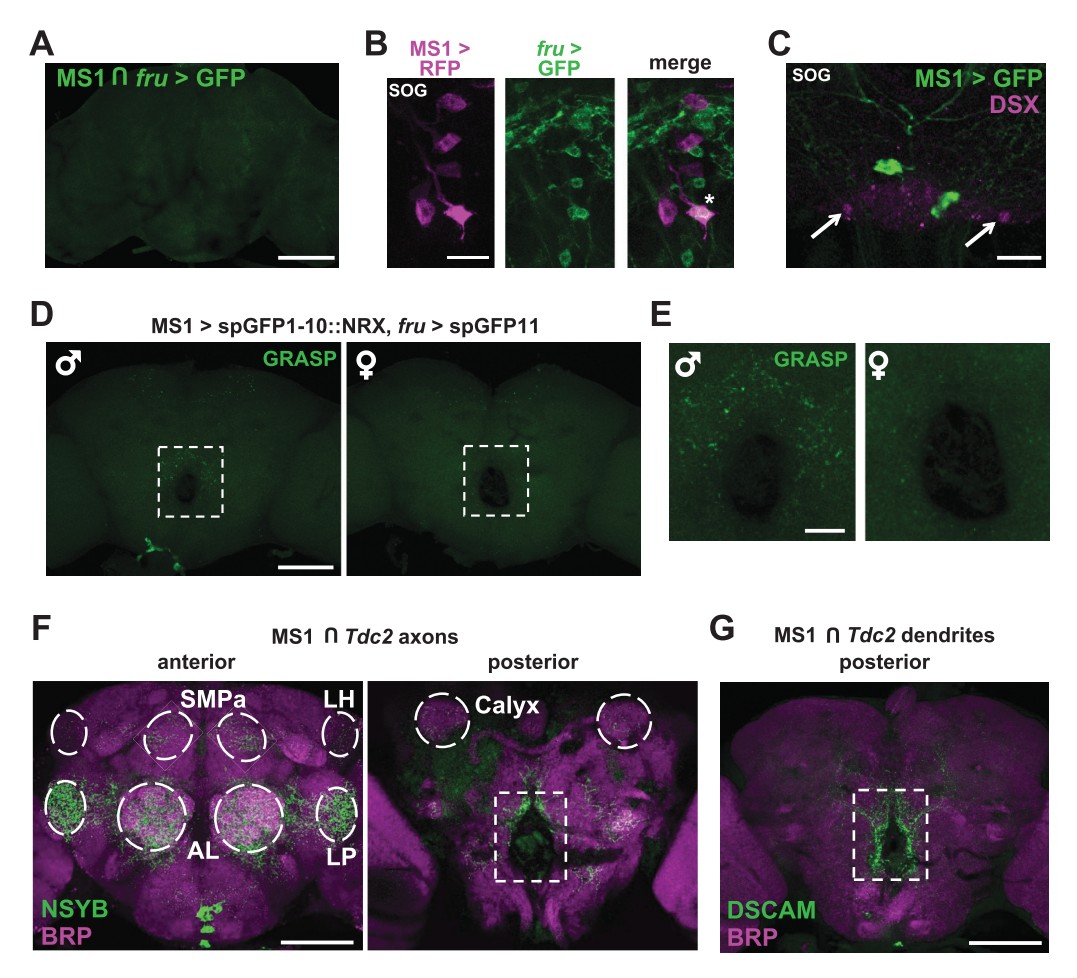

**Figure 5.** MS1 neurons appear to form synaptic contacts with FRU neurons. (**A**) Confocal projection of a male brain expressing CD8::GFP in the intersection between MS1-Gal4 and *fru*-FLP (MS1-Gal4/UAS-FRT-stop-FRT-CD8::GFP; FRU-FLP/+). Scale bar: 100 μm. (**B**) Confocal projection of the SOG region of a male brain expressing RFP (magenta) in MS1 neurons (MS1-Gal4 > UAS-mCD8::RFP) and GFP (green) in FRU neurons (*fru*-LexA > LexAOp-CD8::GFP). *The single MS1 neuron and two FRU neurons are at different focal planes. Scale bar: 25 μm. (**C**) Confocal projection of the SOG region of a male brain expressing CD8::GFP under the control of MS1-Gal4. The brain was co-stained using antibodies against GFP and DSX. Arrows point to DSX positive neurons. Scale bar: 25 μm. (**D**) Confocal projection of a male (left) or female (right) brain expressing spGFP1-10::NRX driven by MS1-Gal4 and spGFP11::CD4 driven by *fru*-LexA. Reconstituted GFP fluorescence was detected without antibody staining. Scale bar: 100 μm. A negative control image is presented in *Figure 5—figure supplement 1B*. (**E**) Magnified view of the region indicated by the rectangle in (**D**). Scale bar: 25 μm. (**F**) Confocal projections of male brains expressing the pre-synaptic marker NSYB in the octopaminergic subset of MS1 neurons (MS1-Gal4 > UAS-FRT-stop-FRT-*nSyb*, *Tdc2*-LexA > LexAop-FLP). Axon terminals are visible in antennal lobes (AL), lateral protocerebrum (LP), lateral horn (LH), anterior superior medial protocerebrum (SMPa), mushroom body calyx (Calyx), and the region surrounding the esophagus (indicated by the rectangle). Scale bar: 100 μm. (**G**) Posterior view of a male brain expressing the post-synaptic marker DSCAM in the octopaminergic subset of MS1 neurons (MS1-Gal4 > UAS-FRT-stop-FRT-*Dscam*, *Tdc2*-LexA > LexAop-FLP). DSCAM expression is highly enriched around the esophagus (indicated by the rectangle). Scale bar: 100 μm.

The following figure supplement is available for figure 5:

**Figure supplement 1.** MS1 neurons do not express FRU but likely form synaptic contacts with FRU neurons.

To determine anatomical connectivity between MS1 and FRU neurons, we examined patterns of pre- and post-synaptic markers in MS1 neurons. When the NSYB pre-synaptic marker was targeted to the octopamingeric subset of MS1 neurons, we observed expression in multiple brain areas known to be involved in male reproductive behavior (*Yamamoto and Koganezawa, 2013*; *Zhang et al., 2016*), including the antennal lobe, lateral horn, lateral protocerebrum, anterior

superior medial protocerebrum, and mushroom body regions (*Figure 5F*). Since FRU neurons also send projections to these regions, it is possible that MS1 neurons make synaptic contacts with several clusters of FRU neurons that were not detected by the GRASP technique. In contrast to the presynaptic marker, the DSCAM post-synaptic marker was enriched around the esophagus (*Figure 5G*), where we observed strong GRASP signals, suggesting that MS1 neurons receive input from the FRU circuit through this region. However, the NSYB pre-synaptic marker was also observed in the region surrounding the esophagus (*Figure 5F*), indicating octopaminergic MS1 neurons are positioned to receive and send information through this region of the brain.

## MS1 and FRU neurons are connected functionally

The potential anatomical connection between MS1 and FRU neurons suggest that the two populations may be functionally connected. We first tested whether MS1 neurons act upstream of FRU neurons by expressing ATP-dependent channel P2X2 (which is not normally expressed in the fly) in MS1 neurons and using live imaging of GCaMP6m in FRU neurons to measure calcium fluctuations in dissected brains. Upon application of ATP, a number of FRU neurons exhibited a marked increase in the GCaMP6m signal (*Figure 6A,B*), indicating that MS1 neurons act upstream of several FRU clusters. Based on the position of their cell bodies, one of these clusters appeared to be P1 interneurons in the posterior brain. Using the more restricted R71G01-LexA driver, we confirmed an excitatory input from MS1 to P1 neurons (*Figure 6—figure supplement 1*).

*Figure 6A* shows MS1 neurons influence activity of multiple other FRU subpopulations. One of these bilateral FRU cluster located above the antennal lobes is likely to be mAL, which provides inhibitory input to P1 neurons (*Clowney et al., 2015*; *Kallman et al., 2015*). In addition, although we could not confirm their identity, a neuronal pair in the superior protocerebrum may be FRU-expressing dopaminergic aSP4 neurons (*Figure 6A*), whose activity has recently been shown to reflect mating drive (*Zhang et al., 2016*). Quantification of the GCaMP6m signal in the cell bodies of mAL neurons as well as the arch neuropil region heavily innervated by FRU neurons showed that stimulation of MS1 neurons led to strong calcium responses in males but not in females (*Figure 6B*). Since P1 neurons are not present in females, excitatory input from MS1 to P1 is also male specific. Additional unidentified FRU clusters responded strongly to activation of MS1 neurons, indicating that MS1 neurons provide broad excitatory input to the FRU circuit.

Consistent with the idea that MS1 neurons act upstream of FRU neurons, we found that activation of MS1 neurons in a *fru* mutant background had little effect on male sleep (*Figure 7A*). In addition, MS1 activation in a *dsx* mutant background produced little effect on male sleep (*Figure 7A*), suggesting that MS1 neurons also act upstream of *dsx*-expressing neurons. P1 neurons express both FRU and DSX (*Kimura et al., 2008*), and our data show that they exhibit calcium responses to MS1 stimulation, and thus are a prime candidate for a FRU cluster acting downstream of MS1 neurons to promote wakefulness. Indeed, activation of P1 neurons using a highly restricted P1-split-Gal4 driver (*Inagaki et al., 2014*) resulted in a strong reduction in male sleep, but had no effect on female sleep as expected from the sexually dimorphic nature of P1 (*Figure 7B*).

Our data suggest P1 neurons act downstream of MS1 neurons to mediate sleep regulation. Artificial activation of P1 has been shown to trigger courtship behaviors in isolated males (*Kohatsu et al., 2011*; *von Philipsborn et al., 2011*), and similarly we observed wing extension in males with activated P1 neurons (*Video 3*). However, wing extension did not last beyond the first hour of thermogenetic activation of P1 neurons, and continued activation led to behavior typical of awake flies such as pacing, feeding, and grooming (*Video 4*). In contrast, MS1 activation did not produce wing extension at any time (*Videos 2–4*), perhaps because activation of P1 via MS1 is not strong enough to trigger courtship behavior. Our data suggest MS1 neurons affect mating success indirectly by providing octopamingeric arousal input to the courtship circuit.

Based on the role of MS1 neurons in female-induced male sleep suppression and the role of FRU neurons in sensory processing of female presence, we hypothesized that MS1 neurons also act downstream of FRU neurons. Indeed, we found that when we used P2X2 and ATP to active the FRU circuit, MS1 neurons in the SOG exhibited an increased GCaMP6m signal in a male-specific fashion (*Figure 7C,D*). We did not detect calcium responses in MS1 neurons when we stimulated P1 neurons using R71G01-LexA (max $\triangle F/F^0$ = 0.01 ± 0.01), which suggests that MS1 and P1 neurons do not form a positive feedback loop and FRU neurons other than the P1 cluster act upstream of MS1 neurons. Collectively, our data show that MS1 neurons receive input from FRU neurons and, in turn,

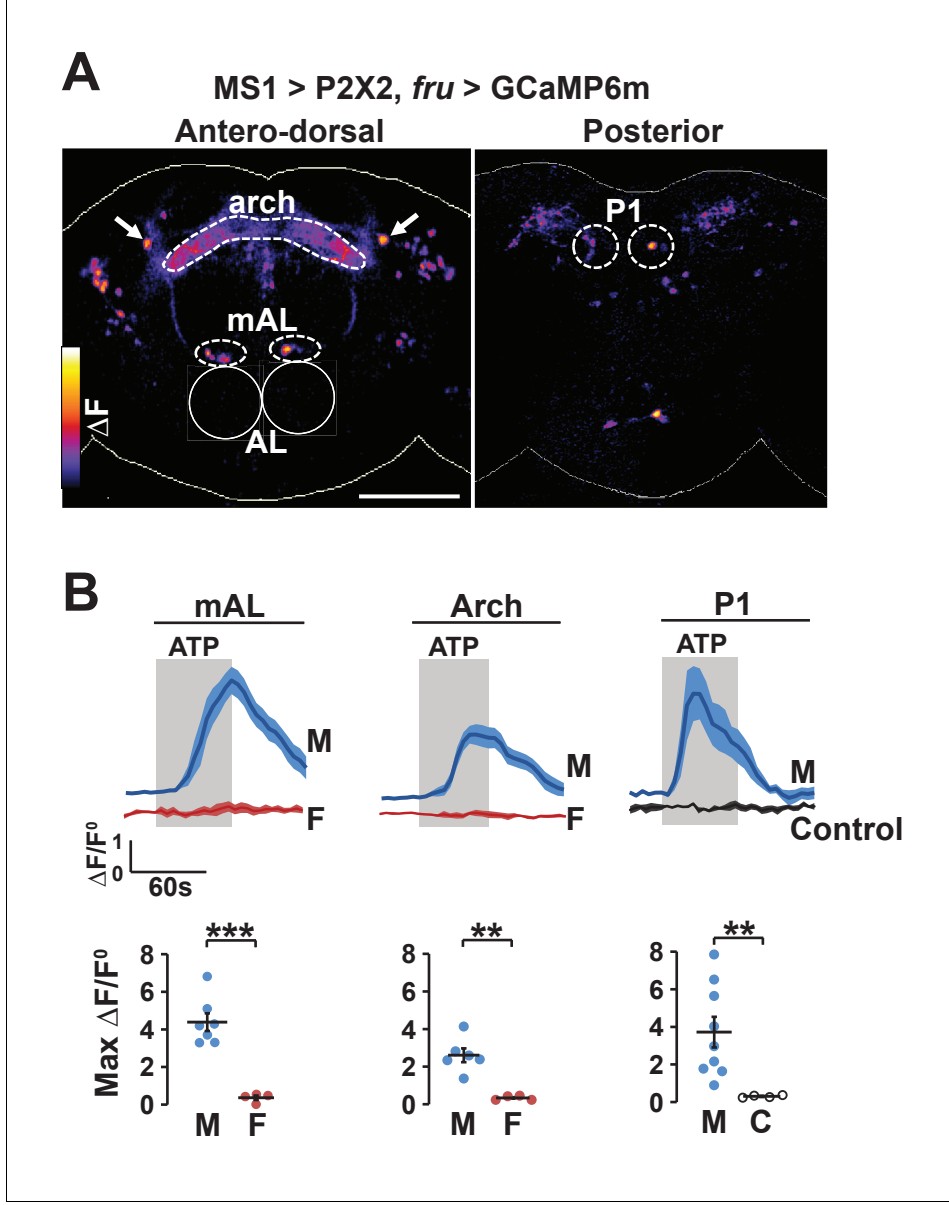

**Figure 6.** Experimental activation of MS1 neurons elicits calcium responses in FRU neurons. (**A**) GCaMP6m increase ($\triangle$F) in FRU neurons of a dissected male brain in which MS1 neurons are activated by P2X2 expression and ATP perfusion. Antero-dorsal (left) and posterior (right) views are presented using the 'fire' look-up table. AL: antennal lobes. Arrows point to a pair of neurons that may be aSP4. Scale bar: 100 µm. (**B**) Normalized GCaMP6m response ($\triangle$F/F$^0$) in the cell bodies of mAL and P1 neurons, and the arch region in male (M), female (F), or negative control (C) brains. Female brains do not exhibit calcium responses in P1 neurons because these neurons are male specific. Flies carrying UAS-P2X2, *fru*-LexA, and LexAop-GCaMP6m, but not MS1-Gal4 served as negative controls. Fluorescence traces (top) and peak responses (bottom) are presented. Gray rectangles indicate 2.5 mM ATP perfusion. N = 4–9. Student's *t* test (**B**).

The following figure supplement is available for figure 6:

**Figure supplement 1.** MS1 stimulation induces calcium responses in P1.

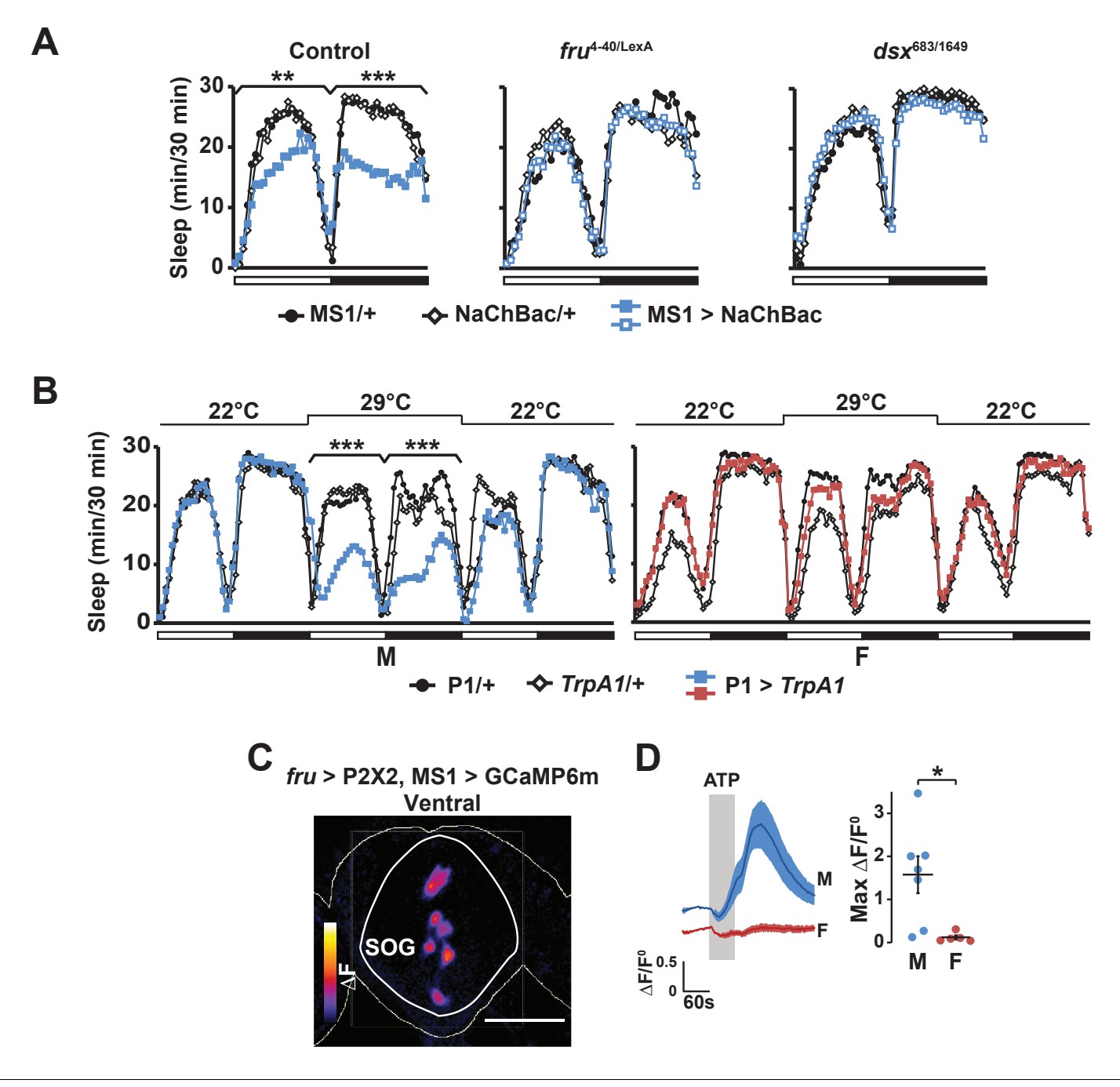

**Figure 7.** Activation of P1 neurons suppresses male sleep and experimental activation of FRU neurons elicits calcium responses in MS1 neurons. (**A**) Sleep profile of males expressing NaChBac in MS1 neurons (MS1>NaChBac) and control males (MS1/+ and NaChBac/+) in an iso31 control (left), *fru*[LexA/4-40] mutant (middle) and *dsx*[683/1649] mutant (right) background. N = 11–83. (**B**) Sleep profile of male (M) and female (F) flies in which P1-split-Gal4 (*Inagaki et al., 2014*) was used to drive UAS-*TrpA1* (P1 > *TrpA1*) and parental control flies (P1/+ and *TrpA1*/+). TrpA1 was activated on the 2nd day by raising the temperature from 22°C to 29°C. N = 82–91. (**C**) GCaMP6m response to ATP (△F) in MS1 neurons of a male brain in which FRU neurons are activated by P2X2 expression and ATP perfusion. (**D**) Normalized GCaMP6m response (△F/F[0]) in the cell bodies of MS1 neurons in males (M) and females (F). N = 5–7. One-way ANOVA followed by Dunnett post hoc tests (**A, B**); Student's *t* test (**C, D**).

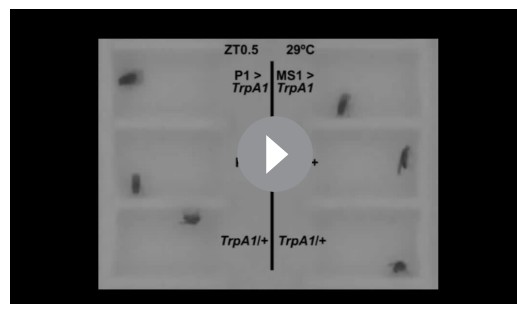

**Video 3.** Male flies with activated P1 neurons and control males (left) as well as male flies with activated MS1 neurons and control males (right) at 29°C. The flies were recorded ~30 min after the switch from 22°C to 29°C (ZT0.5) under white light. Wing extension is seen in males with activated P1 neurons (top left), but not in males with activated MS1 neurons (top right).

send excitatory input to several FRU clusters to balance sleep and sexual behaviors specifically in male flies.

## Discussion

Behavioral choice is a continual challenge facing organisms with multiple goals. Here we have investigated the choice between two essential behaviors: sleep and sex. We found that male flies suppressed sleep in the presence of females, and that sexual satiety or elevated levels of sleep drive attenuated female-induced male sleep suppression. These findings demonstrate that sleep and sex drives compete to control behavior, highlighting the importance of motivational factors such as sex drive in sleep regulation.

A number of wake-promoting neuronal populations in flies and mammals have previously been identified (*Afonso et al., 2015*; *Brown et al., 2012*; *Crocker et al., 2010*; *Joiner et al., 2006*; *Liu et al., 2012*; *Parisky et al., 2008*; *Pitman et al., 2006*; *Saper et al., 2010*; *Sitaraman et al., 2015*; *Ueno et al., 2012*; *Weber and Dan, 2016*), but why activation of these neurons leads to wakefulness is largely unclear. Since sleep is incompatible with many other behaviors, it is plausible that the role of some arousal centers is to keep animals awake so that they can address other pressing needs. For instance, dopaminergic ventral tegmental area (VTA) neurons in mice are required for maintaining wakefulness in the presence of motivating stimuli such as food and sexual partners (*Eban-Rothschild et al., 2016*), and LEUCOKININ-expressing neurons in *Drosophila* promote wakefulness under starvation conditions (*Murakami et al., 2016*). Our research identified a small number of octopaminergic neurons in the SOG that regulate male sleep specifically in a sexual context. MS1 neurons act in concert with the FRU circuit to promote wakefulness and courtship, suggesting that activation of MS1 neurons tips the balance in favor of courtship over sleep. The selective advantage of being able to inhibit sleep drive in a sexual context is demonstrated by our finding that inhibition of MS1 neurons places male flies at a disadvantage when they must compete for sexual partners.

Whereas total sleep deprivation by external stimulation led to suppression of courtship and wakefulness, partial sleep loss due to activation of MS1 neurons did not lead to rebound sleep. This may be because activation of MS1 neurons mimics self-motivated sleep loss in a sexual context, which allows flies to sleep when sleep drive is sufficiently high, and thus prevents accumulation of excessive sleep drive that leads to rebound sleep. Octopamine signaling may inhibit accumulation or expression of sleep drive (*Seidner et al., 2015*), and thus may be especially well suited for adaptive, self-motivated sleep loss under conditions where wakefulness is required for something important such as sex and food (*Siegel, 2012*). Consistent with this view, octopamine mediates starvation-induced foraging behavior (*Yang et al., 2015*). The noradrenergic system in humans, which is similar to the *Drosophila* octopaminergic system, may function in an analogous manner to allow important motivational factors such as sex drive, fear, and hunger to overcome sleep drive.

In addition to promoting wakefulness, octopamine regulates other behaviors such as aggression (*Hoyer et al., 2008*; *Zhou et al., 2008*), choice between courtship and aggression (*Certel et al., 2007*), memory formation

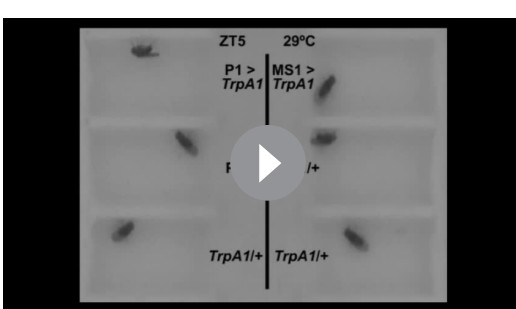

**Video 4.** Flies shown in *Video 3* at ~ZT5 at 29°C. Males with activated P1 neurons no longer exhibited wing extension, but engaged in behavior typical of awake flies such as pacing, feeding, and grooming.

(*Burke et al., 2012*), egg laying (*Monastirioti et al., 1996*), and foraging (*Yang et al., 2015*). MS1 neurons in the SOG are distinct from previously characterized octopaminergic neurons, and play a novel role as a link between sleep and courtship circuits. Each of the behaviors modulated by octopamine may be mediated by distinct subsets of octopaminergic neurons.

Earlier research documented sex differences in sleep (*Isaac et al., 2010*; *Krishnan and Collop, 2006*), yet little has been known about the neural mechanisms underlying sexually-dimorphic regulation of sleep. Enhanced neuronal activity in a subset of dorsal clock neurons (DN1s) was proposed to underlie elevated siesta in males relative to females (*Guo et al., 2016*), but DN1 activation has similar effects on sleep in males and females. The MS1 neurons are unusual in regulating sleep only in males. The sexual dimorphism does not appear to stem from differences in MS1 neurons themselves, but rather from sexually dimorphic connectivity between MS1 and FRU neurons at both anatomical and functional levels. We found that MS1 stimulation elicits calcium responses in several FRU clusters specifically in males. Two of the clusters, P1 and mAL, play important roles in courtship (*Clowney et al., 2015*; *Kallman et al., 2015*; *Kimura et al., 2008*; *Koganezawa et al., 2016*; *Kohatsu et al., 2011*; *Kohatsu and Yamamoto, 2015*). An additional pair in the superior protocerebrum, which may be aSP4 neurons, has been shown to signal mating drive (*Zhang et al., 2016*), raising the possibility that MS1 neurons modulate mating drive depending on social context. A widespread increase in the excitability of the FRU circuit may keep males in a sexually aroused state and provide enhanced sensitivity to cues from females.

In addition to providing excitatory input to several FRU clusters, MS1 neurons receive excitatory input from the FRU circuit. Since MS1 neurons are important for male sleep regulation in the presence of a female especially in the dark, and many FRU neurons respond to female pheromones directly or indirectly, it is plausible that the message conveyed to MS1 neurons from FRU neurons concerns female pheromones. The specific neuronal groups that communicate directly with MS1 neurons have yet to be identified. Nevertheless, MS1 neurons are well positioned to translate the detection of female cues into an arousal signal for sustained courtship. A heightened state of arousal may be especially important for successful mating when sleep drive is high and vision is limited, conditions under which MS1 activity strongly impacts sleep and courtship.

Our work demonstrates that sex drive and sleep drive are integrated in a circuit that contains optopaminergic neurons and FRU neurons, and provides a valuable entry point for investigating the neural circuitry underlying the coordination of sleep and courtship, and more generally the choice between competing behaviors.

## Materials and methods

### Fly stocks

Flies were raised on standard food containing molasses, cornmeal, and yeast under a 12 hr:12 hr light:dark cycle. MS1-Gal4 (BDSC#12837) (*Bellen et al., 2004*), *Tdc2*-LexA (BDSC#52242) (*Shearin et al., 2013*), UAS-*TrpA1* (BDSC#26263) (*Hamada et al., 2008*), UAS–m*CD8::GFP* (BDSC#5137) (*Lee and Luo, 1999*), lexAop2-mCD8::GFP, UAS-IVS-*mCD8::RFP* (BDSC#32229), UAS-*NaChBac::eGFP* (BDSC# 9466) (*Luan et al., 2006*), UAS-*TNT* (BDSC#28838) (*Sweeney et al., 1995*), LexAop2-*FLPL* (BDSC#55820), GMR23E10-Gal4 (BDSC#49032), GMR71G01-lexA (BDSC#54733), UAS-*GCaMP6m* (BDSC#42750), and iso31 ($w^{1118}$) (BDSC#5905) were obtained from the Bloomington Stock Center. UAS-FRT-stop-FRT-*mCD8::GFP*, UAS-FRT-stop-FRT-*Dscam::GFP* and UAS-FRT-stop-FRT-*nSyb::GFP* and *fru-FLP* (*Yu et al., 2010*), LexAop-*GCaMP6m* and UAS-*P2X2* (*Lima and Miesenböck, 2005*), LexAop-*P2X2* (*Yao et al., 2012*), and *fru*[4-40] (*Demir and Dickson, 2005*) lines were obtained from Barry Dickson; *dsx*[683-7058] and *dsx*[1649-9625] mutants (*Chatterjee et al., 2011*), and *fru*-LexA (*Mellert et al., 2010*) from Bruce Baker; LexAop-Gal80 (*Thistle et al., 2012*) and LexAop-*spGFP11::CD4* (*Gordon and Scott, 2009*) from Kristin Scott; UAS- *spGFP1-10::NRX* (*Fan et al., 2013*) from Nirao Shah; P1-split Gal4 (*Inagaki et al., 2014*) from David Anderson; *Tbh*[nm18] mutants (*Monastirioti et al., 1996*) from Maria Monastirioti; and *Oamb* [286] mutants (*Lee et al., 2003*) from Kyung-An Han. Fly lines used in behavioral experiments were outcrossed to an isogenic background (iso31) for at least five generations, except for the *Tbh* and *Oamb* lines.

## Sleep analysis

For sleep analysis, 4- to 7-day-old flies entrained to a 12 hr:12 hr LD cycle were placed in glass tubes containing 5% sucrose and 2% agar. Flies were raised and monitored at 25°C except where noted. Males and females were housed together in groups of ~30 flies until they were loaded into tubes. For experiments involving TrpA1, flies were raised in LD at 22°C and monitored for 1 day at 22°C to determine baseline levels, 1 day at 28–29°C to activate the TrpA1 channel, and 1 day at 22°C to examine recovery. For TNT experiments comparing MM and MF pairs, flies were raised and assayed at 22°C because initial data suggested that UAS-TNT/+ controls behaved differently from other controls at 25°C, perhaps due to leaky TNT expression. Activity data were collected in 1 min bins using *Drosophila* Activity Monitoring (DAM) System (Trikinetics, Waltham, MA). Beam breaks from single-beam (SB) monitors with infrared (IR) detectors at a single location or inter-beam movements from multi-beam (MB) monitors with IR detectors at 17 locations (*Garbe et al., 2015*) were used to measure sleep as a period of inactivity lasting at least 5 min (*Huber et al., 2004*). SB monitors were used for all experiments involving isolated flies, while both SB and MB monitors were used in the MF interaction experiments. We found that although the absolute sleep levels were somewhat higher with SB monitors, the same pattern of reduced sleep in MF compared to MM pairs was seen using SB or MB monitors. For video recording, flies were loaded into 7 mm x 16 mm x 3 mm recording arenas. For nighttime recording, a USB webcam (Logitech Webcam Pro 9000) and infrared LEDs were used as previously described (*Faville et al., 2015*), and for daytime recording, a digital camera (Sony DCR-SX63) and white LEDs were used. For sleep deprivation experiments, flies placed in SB monitors or recording arenas were deprived of sleep using mechanical stimulation. A multi-tube vortexer fitted with a mounting plate (Trikinetics, Waltham, MA) was used to apply mechanical stimulation for 3 s every min. Satiety manipulation was essentially as described (*Zhang et al., 2016*), except that single virgin males were grouped with 10–15 virgin females for 4.5–5.5 hr to induce satiety. Mating behavior (courtship and copulation) was scored at the beginning and end of the satiety assay, which confirmed that males were satiated by the end of the assay. Immediately following the satiety manipulation, shortly before ZT12, individual male flies were aspirated into monitor tubes or recording arenas that contained control females for sleep assay in the MF condition. For DAM data, sleep parameters were analyzed using a MATLAB-based software, SleepLab (William Joiner). For video data, sleep amount of individual flies was manually scored for the first 5 min of each nighttime hour, except for the sleep deprivation experiment, where the first 5 min of each 30 min interval during 6 hr after deprivation was scored. Scoring was done blind to the experimental condition and genotype. In cases where only one fly in a MF pair was active, we used male courtship behavior to determine its sex.

## Analysis of mating behavior

For simultaneous analysis of courtship and sleep during the night, videos recorded under infrared light were manually scored for courtship and sleep during 5 min periods as indicated. For analysis of courtship during the day, virgin male flies were collected and housed in groups of ~10 on standard fly food for 4–8 days. Iso31 virgin females (3–7 days post-eclosion) were used in non-competitive assays, which were performed during the day phase (ZT1-6). For non-competitive courtship assays, a male and female were gently aspirated into a plastic mating chamber (15 mm diameter and 3 mm depth) covered with a clear plastic plate, and were kept separated until a divider was removed after ~10 min. For the light condition, dim light (~25 lux) was used because bright light has been shown to interfere with courtship performance of *white* males (*Krstic et al., 2013*). For the dark condition, infrared light was used. Flies were recorded for 90 min using a USB webcam (Logitech Webcam Pro 9000) and scored blind to experimental condition. Courtship index was determined as the fraction of total time a male was engaged in courtship activity during a period of 5 min or until successful copulation after courtship initiation. Courtship activity included orienting, chasing, singing, attempted copulation, and 'scanning', a behavior specific to the dark condition, where the male extends both wings in search of a female (*Krstic et al., 2009*). Only males that took at least 1 min to copulate after courtship initiation were included in the computation of courtship index. Competitive copulation assays were performed under dim red light using Canton-S virgin females. Males of different genotypes were marked with a small dot of acrylic paint on their thorax. Two males and one female were aspirated into each well of a 12-well plate, and the first male to successfully copulate

within 90 min was determined the winner. Trials in which neither male succeeded in copulating were not included in the analysis.

## Immunohistochemistry and GRASP

For whole mount immunostaining, fly brains were fixed in 4% paraformaldehyde (PFA) for 30 min, dissected, and blocked in 5% normal goat serum for 1 hr at RT. The following primary antibodies were used: rabbit anti-GFP (Molecular Probes, Eugene, OR, Cat# A-21312, RRID:AB_221478) at 1:500, mouse anti-RFP (Rockland Cat, Limerick, PA, # 600-401-379, RRID:AB_2209751) at 1:500, BRP (DSHB, Iowa City, IA, Cat# nc82, RRID:AB_528108) at 1:150; anti-HA (Covance Research Products Inc, Princeton, NJ, Cat# MMS-101R-500, RRID:AB_10063630) at 1:1000; and anti-DSX (kind gift from Bruce Baker) at 1:300. The secondary antibodies, Alexa Fluor 488 goat anti-rabbit (Thermo Fisher Scientific, Waltham, MA, Cat# A11008, RRID:AB_143165) and Cy5 goat anti-mouse (Thermo Fisher Scientific, Waltham, MA, Cat# A10524, RRID:AB_2534033) were used at 1:1000. Primary and secondary antibodies were incubated at 4°C overnight. For GRASP experiments, fly brains were fixed in PFA for 30 min at RT, and imaged without immunostaining. Images were obtained on a Leica SP8 confocal microscope.

## Calcium imaging

4- to 7-day-old flies that were housed individually and entrained to LD cycles were anesthetized on ice and dissected in adult hemolymph-like saline (AHL [*Wang et al., 2003*]), and brains were mounted on a glass-bottom chamber containing AHL. A custom-built gravity-dependent perfusion system was used to control perfusion flow. Leica SP8 confocal microscope was used to acquire 20 to 25 slices (~2.5 µm/slice) of the antero-dorsal, ventral, or posterior brain every 2.5 or 5 s for 3 to 5 min. 2.5 mM ATP in AHL was delivered for 1 min after 1 min of baseline measurements. FIJI was used to compute projections of relevant confocal slices, and regions of interest (ROIs) were selected using images taken at high laser intensity. The average intensity of the ROIs during the 30s period before the start of ATP perfusion was used as the baseline measurement, $F^0$. For each time point, normalized $\triangle F$, $(F-F^0)/F^0$, was computed.

## Statistical analysis

To compare multiple groups, one-way ANOVAs were performed followed by Tukey or Dunnett post-hoc tests. Two-way ANOVAs were performed to test for the interaction in experiments involving two factors. Student's *t* tests were used to compare pairs of groups. Log-rank tests were used for cumulative courtship initiation rate and cumulative copulation success rate in non-competitive mating assays. For competitive copulation data, binomial tests were used to assess whether the observed percentage was different from 50%. Bonferroni corrections were applied to correct for multiple tests performed on data from the same flies (e.g., daytime and nighttime). All experiments were repeated on at least two separate occasions using flies from independent genetic crosses, and pooled data are presented.

## Acknowledgements

We thank David Anderson, Bruce Baker, Barry Dickson, Kyung-An Han, Maria Monastirioti, Kristin Scott, and the Bloomington Stock Center for fly stocks; Bruce Baker for DSX antibody; Benjamin Kottler for setting up the DART system; Paula Haynes for advice on calcium imaging; William Joiner for the SleepLab software; Huihui Pan and Victoria Baccini for technical assistance; and Jennifer Wilson, and Amita Sehgal and Matthew Dalva for comments on the manuscript.

# Additional information

## Funding

| Funder | Grant reference number | Author |
|---|---|---|
| National Institute of Neurological Disorders and Stroke | R21NS094782 | Kyunghee Koh |

| National Institute of Neurological Disorders and Stroke | R01NS086887 | Kyunghee Koh |
| National Heart, Lung, and Blood Institute | T32HL07713 | Emilia H Moscato |
| Burroughs Wellcome Fund | Career Award for Medical Scientists | Matthew Kayser |
| Portuguese Foundation for Science and Technology | SFRH-BD-52321-2013 | Daniel R Machado |
| Portuguese Foundation for Science and Technology | SFRH-BD-51726-2011 | Dinis JS Afonso |

The funders had no role in study design, data collection and interpretation, or the decision to submit the work for publication.

### Author contributions

DRM, Conceptualization, Data curation, Formal analysis, Funding acquisition, Writing—original draft; DJSA, Conceptualization, Data curation, Formal analysis, Funding acquisition, Writing—review and editing; ARK, AÖr-Ç, BM, Data curation, Formal analysis, Writing—review and editing; EHM, Data curation, Formal analysis, Funding acquisition, Writing—review and editing; MK, Data curation, Formal analysis, Supervision, Funding acquisition, Writing—review and editing; KK, Conceptualization, Data curation, Formal analysis, Supervision, Funding acquisition, Writing—original draft

### Author ORCIDs

Daniel R Machado, http://orcid.org/0000-0002-4792-2956
Dinis JS Afonso, http://orcid.org/0000-0002-7296-0532
Kyunghee Koh, http://orcid.org/0000-0003-0847-8204

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
