## [Decision Letter]

[Editors’ note: this article was originally rejected after discussions between the reviewers, but the authors were invited to resubmit after an appeal against the decision.]

Thank you for submitting your work entitled "A neural circuit underlying sleep suppression by male sex drive" for consideration by *eLife*. Your article has been reviewed by three peer reviewers, one of whom, Lesie Griffith (Reviewer #1), is a member of our Board of Reviewing Editors, and the evaluation has been overseen by a Senior Editor.

Our decision has been reached after consultation between the reviewers. Based on these discussions and the individual reviews below, we regret to inform you that your work will not be considered further for publication in *eLife*.

The control of the homeostat is incredibly important and is poorly understood. Unfortunately, there were substantial concerns about the quality of the behavioral data due to the methods used to assess behavior that preclude publication in *eLife* at this time. The major issues included:

1) Using the DAM system to assess courtship is not adequate. It is impossible to know who is moving much less why they are moving. There are many methods available now for the detailed assessment of complex behaviors. To make the claims the authors make about motivation and courtship, they must actually measure courtship.

2) It does not seem likely that the long term changes in activity reflect courtship. Many labs have noted that courtship is quite a short term state- it either ends in copulation or frustration, but it ends. The long term increase in locomotion observed with MF pairs may not be due to a single behavioral state. This needs some detailed investigation.

3) There is an apparent contradiction between data in Figure 4 that needs to be resolved. If silencing MS1 changes sleep, how can it do that w/o changing courtship if the author's model is correct?

While we would hope that more detailed observations of the behavior will be productive, the revisions necessary to substantiate your model and to fully understand the courtship aspects of this phenomenon would require more work than would be considered suitable for a "revision".

*Reviewer #1:*

This paper shows very nicely that males specifically suppress sleep to undertake courtship of females. The circuit described in this paper is novel and interesting. The ability to prioritize sleep as part of a hierarchy of behaviors has been reported, but this is the first cellular mechanism proposed. There are also a few interesting tidbits that suggest that this circuit may interact with the homeostat This report (and more directly the accompanying paper) give us a foothold that may allow better understanding of the homeostat.

The one concern I have with the data is the use of the DAM system. You cannot know what or who is moving or why. There is also a point that is confusing which the authors might want to spend some time either clarifying or at least discussing having to do with interpretation of results in light of the accompanying paper from the Gilestro lab. This paper shows that sleep deprivation can suppress courtship. The accompanying paper suggests the opposite hierarchy. I realize that the experiments are not done in exactly the same way- in this paper the authors are looking at the actual courtship and find that sleep takes priority. In the Gilestro paper they are looking at rebound in the absence of a female and find that prior courtship (residual pheromone?) suppresses rebound. This kind of makes no sense- the male is willing to give up an actual female to sleep but will not sleep if there is just residue? This should be addressed in both papers.

*Reviewer #2:*

Machado and Afonso et al. identify a small population of octopaminergic neurons that suppress sleep only in male flies and argue that these neurons are involved in the interaction between mating and sleep drives. There are a couple of potentially interesting leads in here, but overall it is of very low quality and confusing.

The authors claim that the presence of a female fly increases male sexual arousal and therefore reduces male sleep. The data does not come anywhere close to proving this relatively simple behavioral point. Male sleep of the genotype in question is never actually measured since the monitors cannot distinguish between the activity of the two flies (the genotype and the other wild type fly). Is it not possible that the increased activity comes from the female in MF pairs? And why are we not shown what would happen with FF pairs? Even if the male is really staying awake in MF pairs in a way that the females in FF or MF pairs would not, the idea that mating drive is responsible is highly questionable. The authors show a video of courtship, which proves nothing more than that courtship is possible between male and female flies. But do the authors really think the male is courting the female all night long? This seems unlikely: pairs of flies either mate within the first 30 minutes and then largely ignore one another; or don't mate and then ignore each other. The lack of involvement of mating drive in the circuitry studied here is also clear from the data in the paper, where silencing MS1 has no effect on courtship (Figure 4), yet increases sleep in the male-female pair (Figure 4).

The authors use fru and *dsx* mutants (both of whose gene products are required for development) to "confirm… sleep loss requires appropriate male sexual behavior" (Figure 2). One could make the same argument by removing 2 of the flies 6 legs, which would also lead to an increase in "nighttime sleep" and show a critical role for those legs in the sleep/mating drive interaction. This is very weak evidence due to the developmental confound.

The role of MS1 in mating drive, if any, is not clear. Activating MS1 has no effect on courtship (Figure 4). Silencing MS1 decreases mating success in courtship competition assays (Figure 4), but this could easily come from a defect in attaching to the female, etc. Without strong behavioral phenotypes, the significance of the GRASP signal and functional connectivity between MS1 and ~1500 fru+ neurons is not obvious.

The effects of activating MS1 and P1 on sleep are the most interesting aspect of the manuscript, but are preliminary and confusing. A huge fraction of Gal4 lines (>20% in a screen of flylight lines) change sleep patterns or quantity when stimulated with TrpA1. But P1 and MS1 do so only in males, findings that could possibly lead somewhere interesting. This should be further explored.

*Reviewer #3:*

Sleep is important, but sometimes other behaviors, such as reproduction, are more important and are able to override sleep. Here, the Koh lab explores the phenomenon that male fruit flies show an enormous reduction in sleep when they are paired with a female conspecific. In a set of elegant experiments, Machado et al. identify a subset of male-specific octopaminergic neurons that induce wake in the presence of a female (but not a male). Sleep deprivation or activation of the dorsal fan shaped body, a well-known sleep promotingregion in the fly brain are still able to induce sleep in the presence of a female. Next, Machado et al. use a series of anatomical studies to demonstrate that MS1 neurons communicate with fruitless neurons to detect the presence of a female and induce a state of arousal.

[Editors’ note: what now follows is the decision letter after the authors submitted for further consideration.]

Thank you for submitting your article "A neural circuit underlying sleep suppression by male sex drive" for consideration by *eLife*. Your article has been reviewed by two peer reviewers, and the evaluation has been overseen by Leslie Griffith as the Reviewing Editor and by a Senior Editor. The following individual involved in review of your submission has agreed to reveal their identity: Ravi Allada (Reviewer #3).

The reviewers have discussed the reviews with one another and the Reviewing Editor has drafted this decision to help you prepare a revised submission.

Summary:

This paper addresses an interesting and fundemental aspect of sleep – its integration with other critical behaviors. The authors show that mating drive and sleep drive are integrated by a circuit that contains octopaminergic neurons. This provides an important start for understanding this type of hierarchy. The revised version is substantially improved with the new behavioral analysis providing convincing support to the thesis, but the reviewers suggested several revisions that would bring conclusions more in line with data and make the paper more concise.

Essential revisions:

1) Change the title to something that is more in line with what the data show like "Octopaminergic modulation of mating drive suppresses sleep". The term circuit is used in the current title but there is not really a well-defined circuit, only a very interesting neuron. The rest of the circuit is as of yet undefined- resting among the many fru+ neurons (more below re model).

2) Solidify the video scoring by doing it for some period every hour and being clear about which times they are scoring.

3) Revise or delete the model. It is difficult to come up with a model for what MS1 does that is consistent with all of the data. Some of the phenotypes are really striking, but they often contradict one another, or are inconsistent across similar experiments. The most the authors can say is "MS1 neurons are required for suppression of sleep by courtship drive at night and their artificial stimulation promotes wakefulness". This may make them candidates for neurons that mediate the interaction between these drives, but that is far from proven. So this means that nothing is really known about the circuitry through which sleep is suppressed. This isn't so bad, because it is a hard problem to address, but this makes the model so simplistic as to be obviously wrong. The idea that MS1 neurons stimulate P1 neurons to drive courtship in the presence of a female is just not right, for many reasons, including that silencing MS1 has no courtship drive phenotype. The only two things that are really solid about MS1's behavioral functions are that it suppresses sleep specifically in males, and that it is required for nighttime courtship in the presence of a female.

---

## [Author Response]

[Editors’ note: the author responses to the first round of peer review follow.]

*The control of the homeostat is incredibly important and is poorly understood. Unfortunately, there were substantial concerns about the quality of the behavioral data due to the methods used to assess behavior that preclude publication in eLife at this time. The major issues included:*

*1) Using the DAM system to assess courtship is not adequate. It is impossible to know who is moving much less why they are moving. There are many methods available now for the detailed assessment of complex behaviors. To make the claims the authors make about motivation and courtship, they must actually measure courtship.*

Following the reviewers’ suggestion, we performed several additional experiments (Figure 1, Figure 2, Figure 4, Figure 5). We determined sleep and courtship behaviors of male flies in male-male (MM) or male-female (MF) pairs by video recording them overnight and manually scoring sleep-wake and courtship behaviors. We found that during most of the night, males in MF pairs are awake, and they spend the majority of their wake time in courtship (Figure 1). Moreover, sleep-wake behaviors of pairs of flies are highly synchronized such that the amount of time a pair of flies are asleep (i.e., when both flies are asleep) is a close approximation of the sleep amount of the male in the pair, validating the use of the DAM system to assess male sleep in pairs of flies.

*2) It does not seem likely that the long term changes in activity reflect courtship. Many labs have noted that courtship is quite a short term state- it either ends in copulation or frustration, but it ends. The long term increase in locomotion observed with MF pairs may not be due to a single behavioral state. This needs some detailed investigation.*

Our video analysis revealed that a male paired with a female spends much of the night engaged in courtship (Figure 1). The Gilestro lab obtained similar results using an assay condition similar to ours (personal communication). In both labs, flies are monitored in relatively small spaces. The close proximity presumably allows the male to be readily aroused by female cues, providing us with a special opportunity to investigate the balance between sleep and sex drives.

*3) There is an apparent contradiction between data in Figure 4 that needs to be resolved. If silencing MS1 changes sleep, how can it do that w/o changing courtship if the author's model is correct?*

The apparent contradiction is likely due to differences in assay conditions: sleep was assayed during the night whereas courtship was assayed during the day. In support of this view, when we analyzed nighttime videos of MF pairs, we found that males with silenced MS1 neurons exhibited markedly reduced courtship index (Figure 4), linking the change in sleep with a corresponding change in courtship. On the other hand, when we examined courtship during the day, whereas there was little effect of silencing MS1 neurons on courtship index (Figure 5), it did affect copulation latency in the dark (Figure 5).

Reviewer #1:

*This paper shows very nicely that males specifically suppress sleep to undertake courtship of females. The circuit described in this paper is novel and interesting. The ability to prioritize sleep as part of a hierarchy of behaviors has been reported, but this is the first cellular mechanism proposed. There are also a few interesting tidbits that suggest that this circuit may interact with the homeostat This report (and more directly the accompanying paper) give us a foothold that may allow better understanding of the homeostat.*

*The one concern I have with the data is the use of the DAM system. You cannot know what or who is moving or why.*

This concern is addressed in our response to Major Point #1.

*There is also a point that is confusing which the authors might want to spend some time either clarifying or at least discussing having to do with interpretation of results in light of the accompanying paper from the Gilestro lab. This paper shows that sleep deprivation can suppress courtship. The accompanying paper suggests the opposite hierarchy. I realize that the experiments are not done in exactly the same way- in this paper the authors are looking at the actual courtship and find that sleep takes priority. In the Gilestro paper they are looking at rebound in the absence of a female and find that prior courtship (residual pheromone?) suppresses rebound. This kind of makes no sense- the male is willing to give up an actual female to sleep but will not sleep if there is just residue? This should be addressed in both papers.*

The critical difference may be self-motivated partial sleep loss vs externally imposed total sleep deprivation. We show that externally imposed total sleep deprivation can lead to courtship suppression (Figure 2). In contrast, the Gilestro lab shows that self-motivated partial sleep loss due to female presence does not lead to rebound sleep, presumably because males are allowed to sleep when their sleep drive reaches a critical point. We address this point in Discussion section.

Reviewer #2:

*[…] The authors claim that the presence of a female fly increases male sexual arousal and therefore reduces male sleep. The data does not come anywhere close to proving this relatively simple behavioral point. Male sleep of the genotype in question is never actually measured since the monitors cannot distinguish between the activity of the two flies (the genotype and the other wild type fly). Is it not possible that the increased activity comes from the female in MF pairs? And why are we not shown what would happen with FF pairs? Even if the male is really staying awake in MF pairs in a way that the females in FF or MF pairs would not, the idea that mating drive is responsible is highly questionable. The authors show a video of courtship, which proves nothing more than that courtship is possible between male and female flies. But do the authors really think the male is courting the female all night long? This seems unlikely: pairs of flies either mate within the first 30 minutes and then largely ignore one another; or don't mate and then ignore each other.*

Most of these concerns are addressed by video analysis (see our responses to Major Comments #1 and #2). We found that the increased wakefulness in MF pairs was largely due to male courtship behavior. It was rare (only ~2%) that a female was awake when its male partner was asleep, which indicates that females contribute little to the reduced sleep in MF pairs. We do not show FF pairs because our work is focused on sleep and courtship in males.

*The lack of involvement of mating drive in the circuitry studied here is also clear from the data in the paper, where silencing MS1 has no effect on courtship (Figure 4), yet increases sleep in the male-female pair (Figure 4).*

As noted above in response to Major Comment #3, silencing MS1 did have a major impact on courtship at night (Figure 4). During the day when flies are normally awake, silencing had little effect on courtship index.

*The authors use fru and dsx mutants (both of whose gene products are required for development) to "confirm… sleep loss requires appropriate male sexual behavior" (Figure 2). One could make the same argument by removing 2 of the flies 6 legs, which would also lead to an increase in "nighttime sleep" and show a critical role for those legs in the sleep/mating drive interaction. This is very weak evidence due to the developmental confound.*

We removed Figure 2 because video analysis directly demonstrates that it is the male courtship behavior that keeps MF pairs awake at night.

*The role of MS1 in mating drive, if any, is not clear. Activating MS1 has no effect on courtship (Figure 4). Silencing MS1 decreases mating success in courtship competition assays (Figure 4), but this could easily come from a defect in attaching to the female, etc. Without strong behavioral phenotypes, the significance of the GRASP signal and functional connectivity between MS1 and ~1500 fru+ neurons is not obvious.*

Our new data show that silencing of MS1neurons increases copulation latency in the dark, but not in the light (Figure 5), which suggests that decreased mating success in competitive assays is unlikely to due to the inability to attach to the female or to perform other motor functions necessary for successful copulation. We acknowledge that whether MS1 neurons have a direct role in mating drive is unclear. However, our data showing that MS1 neurons send excitatory input to P1 neurons and that activation of P1 neurons suppresses sleep (together with previous data showing that P1 activation triggers courtship) may explain the observed effects of MS1 manipulation on sleep and courtship. In our revised model we make it explicit that sex drive and female cues cooperate to activate P1 neurons and that MS1 neurons amplify FRU circuit activity to coordinately regulate courtship and sleep.

*The effects of activating MS1 and P1 on sleep are the most interesting aspect of the manuscript, but are preliminary and confusing. A huge fraction of Gal4 lines (>20% in a screen of flylight lines) change sleep patterns or quantity when stimulated with TrpA1. But P1 and MS1 do so only in males, findings that could possibly lead somewhere interesting. This should be further explored.*

We agree that the male-specific sleep regulation by MS1 and P1 is unusual and interesting. We believe our work has made several important discoveries about the neural circuit underlying the balance between sleep and sex drives, a topic that has not been investigated before. We have identified a small group of previously uncharacterized octopaminergic neurons required for sleep loss specifically in a sexual context. It is notable that they do not express FRU or DSX, but make male specific connections with the FRU circuit. The bidirectional functional connectivity between MS1 and FRU neurons suggests a role for MS1 neurons in amplifying FRU circuit activity for sustained sexual arousal. Furthermore, we find that MS1 neurons provide excitatory input to P1 neurons, which we show have a novel role in sleep suppression. For a first paper on the topic, we believe we have made considerable progress.

[Editors’ note: the author responses to the re-review follow.]

*Essential revisions:*

*1) Change the title to something that is more in line with what the data show like "Octopaminergic modulation of mating drive suppresses sleep". The term circuit is used in the current title but there is not really a well-defined circuit, only a very interesting neuron. The rest of the circuit is as of yet undefined- resting among the many fru+ neurons (more below re model).*

We have changed the title to “Identification of octopaminergic neurons underlying sleep suppression by male sex drive”.

*2) Solidify the video scoring by doing it for some period every hour and being clear about which times they are scoring.*

We have performed additional video scoring so that the first 5 minutes of every nighttime hour are scored, except for the sleep deprivation experiment, where the first 5 minutes of every 30-minute interval during 6 hours after deprivation are scored. The results show essentially the same pattern of data as before. We have revised figure legends and methods to clarify which time points are scored.

*3) Revise or delete the model. It is difficult to come up with a model for what MS1 does that is consistent with all of the data. Some of the phenotypes are really striking, but they often contradict one another, or are inconsistent across similar experiments. The most the authors can say is "MS1 neurons are required for suppression of sleep by courtship drive at night and their artificial stimulation promotes wakefulness". This may make them candidates for neurons that mediate the interaction between these drives, but that is far from proven. So this means that nothing is really known about the circuitry through which sleep is suppressed. This isn't so bad, because it is a hard problem to address, but this makes the model so simplistic as to be obviously wrong. The idea that MS1 neurons stimulate P1 neurons to drive courtship in the presence of a female is just not right, for many reasons, including that silencing MS1 has no courtship drive phenotype. The only two things that are really solid about MS1's behavioral functions are that it suppresses sleep specifically in males, and that it is required for nighttime courtship in the presence of a female.*

We agree that further investigation is needed to come up with a satisfactory model, and have thus deleted the model and revised the text accordingly.